# A FREQUENCY DOMAIN ANALYSIS OF GRADIENT-BASED ADVERSARIAL EXAMPLES

## ABSTRACT

It is well known that deep neural networks are vulnerable to adversarial examples. We attempt to understand adversarial examples from the perspective of *frequency analysis*. Several works have empirically shown that the gradient-based adversarial attacks perform differently in the low-frequency and high-frequency part of the input data. But there is still a lack of theoretical justification of these phenomena. In this work, we both theoretically and empirically show that the adversarial perturbations gradually increase the concentration in the low-frequency domain of the spectrum during the training process of the model parameters. And the log-spectrum difference of the adversarial examples and clean image is more concentrated in the high-frequency part than the low-frequency part. We also find out that the ratio of the high-frequency and the low-frequency part in the adversarial perturbation is much larger than that in the corresponding natural image. Inspired by these important theoretical findings, we apply low-pass filter to potential adversarial examples before feeding them to the model. The results show that this preprocessing can significantly improve the robustness of the model.

## 1 INTRODUCTION

Recently, deep neural networks (DNN) have achieved great success in the field of image processing, but it was found that DNNs are vulnerable to some synthetic data called *adversarial examples* ((Szegedy et al., 2013), (Kurakin et al., 2016)). Adversarial examples are natural samples plus *adversarial perturbations*, and the perturbations between natural samples and adversarial examples are imperceptible to human but able to fool the model. Typically, generating an adversarial example can be considered as finding an example in an $\epsilon$-ball around a natural image that could be misclassified by the classifier. Recent studies designed Fast Gradient Sign Method (FGSM, (Goodfellow et al., 2014)), Fast Gradient Method (FGM, (Miyato et al., 2016)), Projected Gradient Descent (PGD, (Madry et al., 2017)) and other algorithms (Carlini & Wagner (2017), (Su et al., 2019), (Xiao et al., 2018), (Kurakin et al., 2016), (Chen et al., 2017)) to attack the model.

Since the phenomenon of adversarial examples was discovered, many related works have made progress to study why they exist. Several works studied this phenomenon from the perspective of feature representation. Ilyas et al. (2019) divided the features into non-robust ones that are responsible for the model's vulnerability to adversarial examples, and robust ones that are close to human perception. Further, they showed that adversarial vulnerability arises from non-robust features that are useful for correct classification.

Another way to characterize adversarial examples is to investigate them in the frequency domain via Fourier transform. Wang et al. (2020a) divided an image into low-frequency component (LFC) and high-frequency component (HFC) and they empirically showed that human can only perceive LFC, but convolutional neural networks can obtain useful information from both LFC and HFC. Yin et al. (2019) filters out the input data with low-pass or high-pass filters to study the sensitivity of the additive noise with different frequencies. Wang et al. (2020b) claimed that existing adversarial attacks mainly concentrate in the high-frequency part. Sharma et al. (2019) found that when perturbations are constrained to the low-frequency subspace, they are generated faster and are more transferable, and will be effective to fool the defended models, but not for clean models.

All of these works showed that spectrum in frequency domain is an reasonable way to study the adversarial examples.

However, there is a lack of theoretical understanding about the dynamics of the adversarial perturbations in frequency domain along the training process of the model parameters. In this work, we focus on the frequency domain of the adversarial perturbations to explore the spectral properties of the adversarial examples generated by FGM (Miyato et al., 2016) and PGD (Madry et al., 2017).

In this work, we give a theoretical analysis in frequency domain of natural images for the adversarial examples.

- For a two-layer neural network with non-linear activation function, we prove that adversarial perturbations by FGM and $l_2$-PGD attacks gradually increase the concentration in the *low-frequency* part of the *spectrum* during the training process over model parameters.

- Meanwhile, the *log-spectrum* difference of the adversarial examples (the definition will be clarified in Section 2.2) will be more concentrated in the *high-frequency* part than the low-frequency part.

- Furthermore, we show that the ratio of the high-frequency and the low-frequency part in the adversarial perturbation is much larger than that in the corresponding clean image.

Empirically,

- we design several experiments on the two-layer model and Resnet-32 with CIFAR-10 to verify the above findings.

- Based on these phenomena, we filter out the high-frequency part of the adversarial examples before feeding them to the model to improve the robustness. Compared with the adversarially trained model with the same architecture, our method achieves comparable robustness with the similar computational cost of normal training and almost no loss of accuracy.

The rest of the paper is organized as follows: In Section 2, we present preliminaries and some theoretical analysis on the calculation of the *spectrum* and *log-spectrum*. Then we provide our main results about the frequency domain analysis of gradient-based adversarial examples in Section 3. Furthermore, some experiments for supporting our theoretical findings are shown in Section 4. Finally we conclude our work and conduct a discussion about the future work in Section 5. All the details about the proof and experiments are shown in the Appendix.

## 2 BACKGROUND

### 2.1 PRELIMINARIES

**Notations** We use $\{0, 1, ..., d\}$ to denote the set of all integers between $0$ and $d$ and use $\| \cdot \|_p$ to denote $l_p$ norm. Specifically, we denote by $\| \cdot \|$ the $l_2$ norm. For a $d$-dimensional vector $\boldsymbol{x}$, we use $x_\mu$ to denote its $\mu$-th component with index starting at 0. For a scalar function $f(\boldsymbol{x})\colon \mathbb{R}^d \mapsto \mathbb{R}$, $\nabla_{\boldsymbol{x}} f$ and $\partial_\mu f$ denote the gradient vector and its $\mu$-th component. We let $\mathrm{sgn}(x) = 1$ for $x > 0$ and $-1$ for $x < 0$. Normal training refers to training on the original training data for learning the optimal weights for the neural networks. $\tilde{\boldsymbol{x}} = \mathcal{F}(\boldsymbol{x})$ denotes the Discrete Fourier Transform (DFT) of $\boldsymbol{x}$.

**Discrete Fourier Transform** The $k$-th frequency component $\tilde{g}[k]$ of one-dimensional DFT of a vector $\boldsymbol{g}$ is defined by

$$\tilde{g}[k] = \mathcal{F}(\boldsymbol{g})[k] = \sum_{\mu=0}^{d-1} g_\mu e^{ik\frac{2\pi}{d}\mu},$$

where $k \in \{-d/2, -d/2 + 1, ..., d/2\}$ if $d$ is even and $k \in \{-(d-1)/2, ..., (d-1)/2\}$ if $d$ is odd. For convenience, we always consider an odd $d$ and one can easily generalize the case to even dimensions. For an integer cut-off (*cutoff frequency*) $k_r \in (0, (d-1)/2)$, the Low Frequency Components (LFC) and High Frequency Components (HFC) of $\tilde{g}$ are denoted by

$$\tilde{g}_l[k] = \begin{cases} \tilde{g}[k] & \text{if } -k_c \le k \le k_c, \\ 0 & \text{otherwise} \end{cases} \qquad \tilde{g}_h[k] = \begin{cases} 0 & \text{if } -k_c \le k \le k_c, \\ \tilde{g}[k] & \text{otherwise}. \end{cases}$$

Then the frequency space $\mathbb{R}^d$ can be decomposed as $S_l \bigoplus S_h$ where $\tilde{g}_l \in S_l$ and $\tilde{g}_h \in S_h$. Let $N_{\tilde{g}} = \sqrt{\sum_{k=-d/2}^{d/2} |\tilde{g}[k]|^2}$, then the ratio of a frequency component with respect to the whole frequency spectrum is denoted by

$$\tau(\tilde{g}[k]) = \frac{|\tilde{g}[k]|^2}{N_{\tilde{g}}^2}$$

and the ratio of LFC is denoted by

$$\tau(\tilde{g} \in S_l) = \sum_{k=-k_c}^{k_c} \frac{|\tilde{g}[k]|^2}{N_{\tilde{g}}^2}.$$

**Setup**   In this paper, we consider a two-layer neural network $f : \mathbb{R}^d \mapsto \mathbb{R}$

$$f(\boldsymbol{x}, \boldsymbol{\theta}) = \sum_{r=1}^{m} a_r \sigma(\boldsymbol{w}_r^T \boldsymbol{x}) \tag{1}$$

where $\sigma$ is an activation function, $\boldsymbol{a} = (a_1, a_2, ..., a_m)^T$ is an $m$-dimensional vector and $\boldsymbol{w}_r$'s are $d$-dimensional weight vectors.

Van der Schaaf & van Hateren (1996) showed that natural images obey a power law in the frequency domain. Therefore, to make our settings closer to widely studied image related tasks, we also assume our input $\boldsymbol{x}$ obey the power law

$$|\tilde{\boldsymbol{x}}[k]| \propto k^{-\alpha}$$

to imitate LFC-concentrated images, where the constant $\alpha \geq 1$ and $|\tilde{\boldsymbol{x}}[0]| = \alpha_0$ is another constant. Besides, we choose the cut-off $k_c$ in such a way that

$$\sum_{k=1}^{k_c} \frac{1}{k^{2\alpha}} + \alpha_0^2 > \sum_{k=k_c+1}^{(d-1)/2} \frac{1}{k^{2\alpha}}, \tag{2}$$

which indicates that $\tau(\tilde{\boldsymbol{x}} \in S_l) > \tau(\tilde{\boldsymbol{x}} \in S_h)$ and can be easily satisfied for a small $k_c$.

**Gradient-based adversarial attack**   For a loss function $\ell$, we find $l_p$-norm $\epsilon$-bounded adversarial perturbations by solving the optimization problem

$$\max_{\|\boldsymbol{\delta}\|_p \leq \epsilon} \ell(f(\boldsymbol{x} + \boldsymbol{\delta}), y),$$

where $(\boldsymbol{x}, y) \in \mathbb{R}^d \times \mathbb{R}$ is $d$-dimensional input and output following the joint distribution $\mathcal{D}$.

To solve the above optimization problem, Goodfellow et al. (2014) proposed FGSM, an attack for an $l_\infty$-bounded adversary, and computed an adversarial example as $\boldsymbol{x}_{\text{adv}} = \boldsymbol{x} + \epsilon \operatorname{sgn}(\nabla_{\boldsymbol{x}} \ell(\theta, \boldsymbol{x}, y))$. Miyato et al. (2016) proposed FGM with an $l_2$-bounded adversary, $\boldsymbol{x}_{\text{adv}} = \boldsymbol{x} + \epsilon \frac{\nabla_{\boldsymbol{x}} \ell(\boldsymbol{x}, y)}{\|\nabla_{\boldsymbol{x}} \ell(\boldsymbol{x}, y)\|_2}$. Madry et al. (2017) introduced PGD which training adversarial examples in an iterative way that $\boldsymbol{x}^{t+1} = \Pi_{\boldsymbol{x}+\mathcal{S}}(\boldsymbol{x}^t + \alpha \operatorname{sgn}(\nabla_{\boldsymbol{x}} \ell(\theta, \boldsymbol{x}, y)))$ or $\boldsymbol{x}^{t+1} = \Pi_{\boldsymbol{x}+\mathcal{S}}(\boldsymbol{x}^t + \alpha \frac{\nabla_{\boldsymbol{x}} \ell(\boldsymbol{x}, y)}{\|\nabla_{\boldsymbol{x}} \ell(\boldsymbol{x}, y)\|_2})$

To simplify the analysis, we make the following assumptions on the normal training process:

**Assumption 1** *There exist $\beta', \beta > 0$ such that $\beta' \leq |\partial \ell / \partial f| \leq \beta$.*

**Assumption 2** *If $\ell = \frac{1}{2}(y - f(\boldsymbol{x}, \boldsymbol{\theta}))^2$, $|\partial \ell / \partial f|$ should be a small quantity such that $|\partial \ell / \partial f| = \epsilon^{1+v}$ if the network is well trained in the end stage of normal training, where $0 < v < 1$.*

**Assumption 3** *There exist $\lambda, \gamma > 0$ such that $0 \leq \sigma' \leq \lambda$ and $0 \leq \sigma'' \leq \gamma$[1].*

**Assumption 4** *$a_r$'s are i.i.d drawn from a distribution[2] and will not be updated during the training.*

---

[1]*e.g.* $\lambda = 1$ and $\gamma = \frac{1}{4}$ if $\sigma = \ln(1 + e^x)$.

[2]*e.g.* Bernoulli distribution.

## 2.2 Exploring the log-spectrum and spectrum

Wang et al. (2020a) considered the *log-spectrum* of an image: $20 * \log(|\mathcal{F}(\boldsymbol{x})|)$, where $\boldsymbol{x}$ is a $2d$-dimensional matrix representing an image. One way to calculate the **"log-spectrum difference of the adversarial examples"** is to consider the difference between the log-spectrum of the adversarial and natural image and as follows,

$$20 * |\log(|\mathcal{F}(\boldsymbol{x}+\boldsymbol{\delta})|) - \log(|\mathcal{F}(\boldsymbol{x})|)| \approx 20 * \left|\log(1 + \frac{|\mathcal{F}(\boldsymbol{\delta})|}{|\mathcal{F}(\boldsymbol{x})|})\right| \approx 20 * \frac{|\mathcal{F}(\boldsymbol{\delta})|}{|\mathcal{F}(\boldsymbol{x})|}. \tag{3}$$

We can observe that this quantity is approximately proportional to the ratio of the perturbation's over the natural image's frequency. Moreover, the average Relative Change in discrete cosine Transforms (RCT) in (Wang et al., 2020b)) approximately equals to it. Fig. 5 empirically shows that this ratio is large for high frequency component but small for low frequency one. However, Van der Schaaf & van Hateren (1996) claimed that the LFC of $\mathcal{F}(x)$ is much bigger than its HFC, and the right side of Eq. 3 is inversely proportional to $\mathcal{F}(x)$. The denominator of the right side of Eq.(3) in low-frequency part is much larger than it in high frequency part, so it may be imprudent to consider the frequency distribution of the adversarial perturbations only based on the ratio in the right side of Eq.(3).

Yin et al. (2019) provided another spectral measurement for adversarial examples. They considered the **"spectrum of the adversarial perturbation"** $|\mathcal{F}(\delta)|$, which directly reflects the distribution of adversarial perturbations in the frequency domain. Unfortunately, they claimed that 'The adversarial perturbations for the normally trained model are uniformly distributed across frequency components' without sufficient empirical and theoretical justification. In this work, we provide a totally different theoretical finding that the adversarial perturbations for the normally trained model are gradually concentrated in the low-frequency domain, further verified by extensive experiments.

Fig. 1 visually shows the calculation methods of the *spectrum of perturbations* $\mathcal{F}(\delta)$ and the *log-spectrum difference of adversarial examples* $|\log(|\mathcal{F}(\boldsymbol{x}+\boldsymbol{\delta})|) - \log(|\mathcal{F}(\boldsymbol{x})| \approx \frac{|\mathcal{F}(\boldsymbol{\delta})|}{|\mathcal{F}(\boldsymbol{x})|}$.

## 3 Main Results

In Section 3.1, we show that, for a well trained two-layer neural network described in Eq.(1) with LFC-concentrated inputs, $l_2$-norm adversarial perturbations generated by FGM (Theorem 1) prefer to concentrate in the low frequency domain. Section 3.2 theoretically proves the HFC-concentration of the log-spectrum difference of adversarial examples. Section 3.3 attempts to demonstrate the effectiveness of masking the HFC of the adversarial examples to improve model's robustness despite that perturbations can concentrate in the low frequency domain. All technical proofs are deferred to Appendix A. Results on $l_2$ PGD perturbations are presented in Appendix B.

In this work, we consider the two-layer neural networks and the loss function $\ell = \frac{1}{2}(y - f(\boldsymbol{x}, \boldsymbol{\theta}))^2$. For any $\boldsymbol{x}$ in this setting, we have

$$\frac{\partial f}{\partial a_r} = \sigma(\boldsymbol{w}_r^T \boldsymbol{x}), \quad \frac{\partial f}{\partial \boldsymbol{w}_r} = a'_r \boldsymbol{x}, \quad \nabla_{\boldsymbol{x}} f = \sum_r a'_r \boldsymbol{w}_r,$$

Figure 1: Visualization of spectrum and log-spectrum for adversarial attacks . Section 3.1 focuses on the center picture $\mathcal{F}(\delta)$ to prove that the LFC ratio of $\mathcal{F}(\delta)$ is increasing. Section 3.2 focuses on the third-line center picture $|\log(|\mathcal{F}(x+\delta)|) - \log(|\mathcal{F}(x)|)|$, and shows that its LFC is smaller than its HFC.

where $a'_r = a_r \sigma'(\boldsymbol{w}_r^T \boldsymbol{x})$ and $\boldsymbol{a}' = (a'_1, a'_2, ..., a'_m)$. At the $(t+1)$-th step of gradient descent in the normal training process of the classifier, weight $\boldsymbol{w}$ is updated as

$$\boldsymbol{w}_r^{(t+1)} = \boldsymbol{w}_r^{(t)} - \eta \frac{\partial \ell}{\partial f}^{(t)} a_r'^{(t)} \boldsymbol{x}. \tag{4}$$

We use $L_x$ and $H_x$ to denote the LFC and HFC of the clean input

$$L_x = \sum_{k=0}^{k_c} |\tilde{\boldsymbol{x}}[k]|^2 \text{ and } H_x = \sum_{k=k_c+1}^{(d-1)/2} |\tilde{\boldsymbol{x}}[k]|^2. \tag{5}$$

## 3.1 Spectral trajectories of $l_2$-norm FGM perturbations

Consider the $l_2$-norm FGM perturbation $\boldsymbol{\delta}$ and its DFT $\tilde{\boldsymbol{\delta}}[k]$

$$\boldsymbol{\delta} = \epsilon \frac{\frac{\partial \ell}{\partial f} \nabla_{\boldsymbol{x}} f}{\|\frac{\partial \ell}{\partial f} \nabla_{\boldsymbol{x}} f\|}, \quad \tilde{\boldsymbol{\delta}}[k] = \epsilon \operatorname{sgn}\left(\frac{\partial \ell}{\partial f}\right) \frac{1}{\|\nabla_{\boldsymbol{x}} f\|} \widetilde{\nabla} f[k], \tag{6}$$

where $\widetilde{\nabla} f[k] = \mathcal{F}(\nabla_{\boldsymbol{x}} f)[k]$. Since the coefficient before $\widetilde{\nabla} f[k]$ in (6) is same for different frequencies $k$, we only need to analyze the spectrum of $\widetilde{\nabla} f[k]$ to investigate the ratio of certain frequency component over the whole frequency spectrum, i.e.

$$\tau\left(\tilde{\boldsymbol{\delta}}[k]\right) = \tau\left(\widetilde{\nabla} f[k]\right). \tag{7}$$

In this subsection, we explore the evolving of $\tau(\tilde{\boldsymbol{\delta}}[k])$ along the normal training process. For a randomly initialized network, the FGM-attack perturbation will not have bias towards either HFC or LFC. We attack this model with FGM at each step of the normal training to study the trajectory of these perturbations in the frequency domain. According to Eq.(4), $\nabla_{\boldsymbol{x}} f$ is updated to the order of $\eta$ at the $(t+1)$-th step as

$$\nabla_{\boldsymbol{x}} f^{(t+1)} = \sum_{r=1}^{m} (1 - \eta_r^{(t)}) a_r'^{(t)} \boldsymbol{w}_r^{(t)} - \tilde{\eta}^{(t)} \boldsymbol{x} + \mathcal{O}(\eta^2),$$

where $\eta_r^{(t)} = \eta a_r \|\boldsymbol{x}\|^2 \frac{\partial \ell}{\partial f}^{(t)} \sigma''^{(t)}(\boldsymbol{w}_r^{(t)T} \boldsymbol{x})$ and $\tilde{\eta}^{(t)} = \eta \frac{\partial \ell}{\partial f}^{(t)} \|\boldsymbol{a}'^{(t)}\|^2$ are composite learning rates for easing our notation. For ReLU activation function $\sigma(x) = \max(0, x)$, we have that $\eta_r^{(t)} = \mathbb{E}_{i \in \{1,2,\dots,m\}}[\eta_i^{(t)}]$, therefore, considering that Softplus has similar shape as ReLU, we derive that $\eta_r^{(t)} = \bar{\eta}^{(t)} + \mathcal{O}(\eta^2)$, where $\bar{\eta}^{(t)} = \eta \|\boldsymbol{x}\|^2 \frac{\partial \ell}{\partial f}^{(t)} \mathbb{E}_{r \in \{1,2,\dots,m\}}\left[a_r \sigma''^{(t)}(\boldsymbol{w}_r^{(t)T} \boldsymbol{x})\right]$. In this way, the $1d$-DFT of $\nabla_{\boldsymbol{x}} f^{(t+1)}$ has the following form,

$$\widetilde{\nabla} f^{(t+1)}[k] = (1 - \bar{\eta}^{(t)}) \widetilde{\nabla} f^{(t)}[k] - \tilde{\eta}^{(t)} \tilde{\boldsymbol{x}}[k]. \tag{8}$$

We are now ready to study the trajectory of $\tau(\tilde{\boldsymbol{\delta}}[k])$ along the training step $t$. Let $\triangle \varphi_k^{(t)}$ denote the difference of phases between $\widetilde{\nabla} f^{(t)}[k]$ and $\tilde{\boldsymbol{x}}[k]$ for frequency $k$ at the $t$-th step. And for ease of notation, denote

$$\triangle \tau_l^{(t)} \triangleq -2 \sum_{k=0}^{k_c} |\widetilde{\nabla} f^{(t)}[k]| \, |\tilde{\boldsymbol{x}}[k]| \cos\left(\triangle \varphi_k^{(t)}\right), \quad \triangle \tau_h^{(t)} \triangleq -2 \sum_{k=k_c+1}^{(d-1)/2} |\widetilde{\nabla} f^{(t)}[k]| \, |\tilde{\boldsymbol{x}}[k]| \cos\left(\triangle \varphi_k^{(t)}\right),$$

where $\tilde{\eta}^{(t)} \triangle \tau_l^{(t)}$ and $\tilde{\eta}^{(t)} \triangle \tau_h^{(t)}$ are approximately changed amounts for LFC and HFC of $|\widetilde{\nabla} f^{(t)}|^2$ at the $t$-th step of the normal training of the network, respectively. Since the network is randomly initialized and the clean inputs $\boldsymbol{x}$ are highly concentrated in the low frequency domain, we always study the case where $\tau(\tilde{\boldsymbol{\delta}}^{(0)} \in S_l) < \tau(\tilde{\boldsymbol{x}} \in S_l)$. We now, taking Eq.(7) into consideration, present our main theorem about $l_2$ norm FGM perturbations as follows.

**Theorem 1 (The spectral trajectory of $l_2$ FGM perturbation)** *During the training process of the two-layer neural network $f(x)$ in (1), the $l_2$ norm FGM adversarial perturbation will change its ratio of LFC, $\tau(\tilde{\boldsymbol{\delta}} \in S_l)$, at the $(t+1)$-th step as follows,*

$$\tau(\tilde{\boldsymbol{\delta}}^{(t+1)} \in S_l) = \tau(\tilde{\boldsymbol{\delta}}^{(t)} \in S_l) + \tilde{\eta}^{(t)} \frac{\triangle \tau_l^{(t)} \tau\left(\tilde{\boldsymbol{\delta}}^{(t)} \in S_h\right) - \triangle \tau_h^{(t)} \tau\left(\tilde{\boldsymbol{\delta}}^{(t)} \in S_l\right)}{\sum_{k=0}^{(d-1)/2} |\widetilde{\nabla} f^{(t+1)}[k]|^2}; \tag{9}$$

*there will be a $t_1$ s.t. $\forall t \geq t_1$, we have*

$$\tau(\tilde{\boldsymbol{\delta}}^{(t+1)} \in S_l) > \tau(\tilde{\boldsymbol{\delta}}^{(t)} \in S_l). \tag{10}$$

**Remark** According to theorem 1, during the normal training process for a randomly initialized network with LFC-concentrated input, there will be a $t_0$ s.t. $\forall t \geq t_0$, we have $\triangle \tau_l^{(t)} > \triangle \tau_h^{(t)}$, and $\tau\left(\tilde{\boldsymbol{\delta}}^{(t+1)} \in S_l\right) > \tau\left(\tilde{\boldsymbol{\delta}}^{(t)} \in S_l\right)$ if $\tau\left(\tilde{\boldsymbol{\delta}}^{(t)} \in S_l\right) < \tau\left(\tilde{\boldsymbol{\delta}}^{(t)} \in S_h\right)$. For ReLU activation function, starting with $\tau(\tilde{\boldsymbol{\delta}}^{(0)} \in S_l) = 1/2 - \zeta$ and $\triangle \tau_l^{(0)} = \triangle \tau_h^{(0)} + b\zeta$, we have $t_0 = \max\left\{0, -\frac{b\zeta}{n\eta\beta'(L_x - H_x)}\right\}$, where $n = \min_{t \in [0,t_0]} \|\boldsymbol{a}'^{(t)}\|^2$. Besides, given such a network trained with at least $t_0$ steps, there will be a $t_1 > t_0$ such that $\triangle \tau_l^{(t_1)} \tau\left(\tilde{\boldsymbol{\delta}}^{(t_1)} \in S_h\right) \geq \triangle \tau_h^{(t_1)} \tau\left(\tilde{\boldsymbol{\delta}}^{(t_1)} \in S_l\right)$ and FGM perturbation will increase its ratio of LFC for all $t > t_1$ no matter whether it is smaller than $1/2$. Finally, if $\tau\left(\tilde{\boldsymbol{\delta}}^{(t_2)} \in S_l\right) > \tau\left(\tilde{\boldsymbol{\delta}}^{(t_2)} \in S_h\right)$ for some $t_2 > t_0$, this relation will hold for any $t \geq t_2$.

### 3.2 HIGH-FREQUENCY CONCENTRATION FOR LOG-SPECTRUM DIFFERENCE OF ADVERSARIAL EXAMPLES

In this subsection, we explore spectral trajectories of log-spectrum difference of $l_2$ FGM adversarial examples

$$R_k^{(t)} = \frac{\left|\widetilde{\nabla f}^{(t)}[k]\right|}{|\tilde{\boldsymbol{x}}[k]|} \propto \frac{\left|\tilde{\boldsymbol{\delta}}^{(t)}[k]\right|}{|\tilde{\boldsymbol{x}}[k]|}$$

along normal training step $t$ as in Section 3.1 to study their HFC-concentration phenomenon. Since $|\widetilde{\nabla f}^{(0)}[k]|$ for a randomly initialized network (1) has no bias towards high frequency or low frequency while $|\tilde{\boldsymbol{x}}| \propto k^{-\alpha}$, we adopt the following setting

$$R_k^{(0)} \propto k^\alpha$$

without loss of generality and present below our result on the log-spectrum difference of the adversarial examples.

**Theorem 2 (Concentration of the log spectrum difference)** *For $l_2$ FGM perturbations of the network in (1), for any $k > k' > 0$ which satisfies that $R_k^{(0)} > R_{k'}^{(0)}$, there exists a $t'$ such that $R_k^{(t)} > R_{k'}^{(t)}$ for all $t < t'$ steps of the normal training. For ReLU activation function, we have*

$$t' = \frac{R_k^{(0)} \cos(\triangle \varphi_k^{(0)}) - R_{k'}^{(0)} \cos(\triangle \varphi_{k'}^{(0)})}{\eta\beta\|\boldsymbol{a}\|^2}, \tag{11}$$

*while if the initialization satisfies that $R_k^{(0)} \cos(\triangle \varphi_k^{(0)}) \leq R_{k'}^{(0)} \cos(\triangle \varphi_{k'}^{(0)})$ then $t' = \infty$.*

The theorem shows that although $\tilde{\boldsymbol{\delta}}$ itself can concentrate in the low frequency domain, it does not concentrate as dense in the low frequency domain as the inputs $\tilde{\boldsymbol{x}}$ which obey the power law. Therefore, the log-spectrum difference (*i.e.* equivalent to spectrum of the ratio of the perturbation to the clean data) of adversarial examples will express a HFC-concentrated phenomenon instead (Fig.1).

### 3.3 MASKING HFC OF THE ADVERSARIAL EXAMPLES TO IMPROVE ROBUSTNESS

In this subsection, we compare the ratio of LFC of the adversarial perturbations $\tau(\tilde{\boldsymbol{\delta}} \in S_l)$ and the ratio of LFC of original images $\tau(\tilde{\boldsymbol{x}} \in S_l)$.

**Theorem 3 (Comparison on the LFC ratio of clean data and perturbations)** *For $l_2$ FGM perturbation of the two-layer neural network in (1), at the $t$-th step of normal training, let $\zeta = \tau(\tilde{\boldsymbol{x}} \in S_l) - \tau(\tilde{\boldsymbol{\delta}}^{(0)} \in S_l) > 0$, then we will have $\tau(\tilde{\boldsymbol{\delta}}^{(t)} \in S_l) < \tau(\tilde{\boldsymbol{x}} \in S_l)$ for all $t > 0$ if the initilization of the netowrk satisfies that*

$$\triangle \tau_l^{(0)} < \tau(\tilde{\boldsymbol{x}} \in S_l)(\triangle \tau_l^{(0)} + \triangle \tau_h^{(0)}). \tag{12}$$

**Remark** When both $\triangle\tau_l^{(0)}$ and $\triangle\tau_h^{(0)}$ are positive, condition (12) states that the ratio of low frequency component of $\triangle\tau_l^{(0)} + \triangle\tau_h^{(0)}$ is smaller than that of clean data, which can be easily satisfied for a randomly initialized network. The ratio of LFC of the perturbations $\delta$ will be less than that of clean data $x$ if the condition Eq.(12) is satisfied for $l_2$ FGM perturbations during the normal training process. Masking the HFC of an adversarial example $x + \delta$ will then erase more amount of the perturbation than that of the clean data which leads to a new "not-perturbed-much" adversarial example. As a result, one can expect the improvement for robustness of the model, as demonstrated in Section 4.3 later.

## 4 EXPERIMENT

In this section, we conduct several experiments to validate our theoretical conclusions.

**Setup.** We use the Resnet-32 (He et al., 2016), a fully connected network as the last layer and cross entropy loss, and train all the parameters on the CIFAR-10 dataset with Adam optimizer (Kingma & Ba, 2014). We use PGD-attack with $\epsilon = 8/255$, 40 iterations and step size $\xi = 4/255$.

We supplement more experiments in Appendix C to fill the gap between our theory and more complicated settings: we use the MSE loss and keep other conditions unchanged to resolve the discrepancy of loss function; experiments to support our theorems on two-layer neural network with fixed outer-layer parameters are also provided.

### 4.1 THE INCREASE OF LFC CONCENTRATION

In this part, we empirically verify the theorem in Section 3.1, to show that adversarial perturbations gradually concentrate more in the low-frequency part of the spectrum during the training process. Consider a neural network with random initialization, each time before updating all the parameters of neural network, we randomly sample 100 images and apply the $l_2$ FGM / PGD attack to them. For each iteration in PGD attack, we calculate their ratios of LFC and plot them in Fig. 2. It shows that the LFC ratios of the perturbations $\tau(\tilde{\delta} \in S_l)$ are gradually increasing along the normal training.

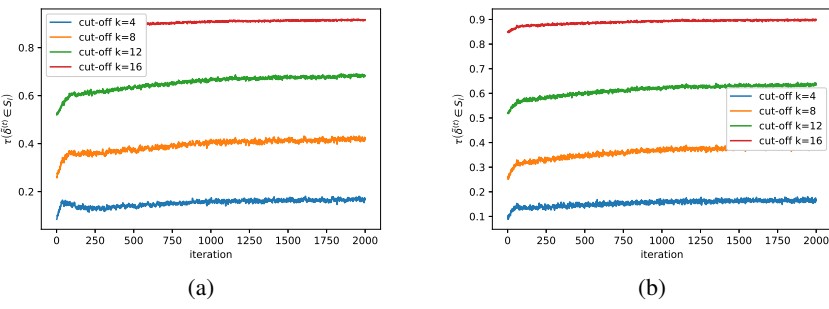

Figure 2: The LFC ratio of the adversarial perturbations in the training process. (a) The attack method is $l_2$ FGM. (b) The attack method is $l_2$ PGD

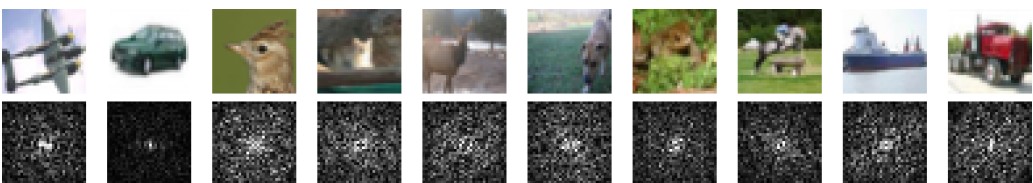

Figure 3: The spectrum of adversarial perturbations. The first line is the original examples in CIFAR-10 dataset. The second line is the spectrum of the adversarial perturbations corresponding to the above images.

For a sufficiently trained neural network, we choose an image, use PGD-attack to obtain its adversarial examples and perturbations, and calculate its spectrum. We randomly select 10 images in the test set and show the original images and the spectrum of perturbations in Fig. 3. We also randomly select 1000 images in the test set and average their spectrum to show the expectation of the spectrum in Fig. 4(a). They show that the spectrum of adversarial perturbations are more concentrated in the low-frequency domain after a sufficient training. Ortiz-Jimenez et al. (2020) show that the distance to the boundary (margin) in different frequency bands is heavily dependent of the distribution used for training, which also supports our claims.

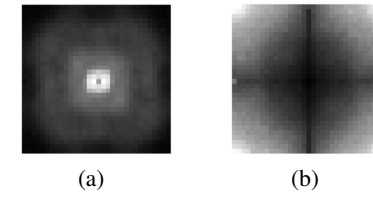

(a)        (b)

Figure 4: The expectation of: (a) the spectrum of adversarial perturbations; (b) the log-spectrum difference of adversarial examples.

## 4.2 HFC CONCENTRATION OF LOG-SPECTRUM DIFFERENCE

We first use PGD-attack to get the adversarial example of a image for a sufficiently trained neural network. We then make the log-spectrum of the original image and its adversarial example and calculate the difference of their log-spectrum (Fig.5). Fig.4(b) displays the expectation of the spectrum for 1000 random images in the test set. They both show that the log-spectrum difference of adversarial examples is generally concentrated around.

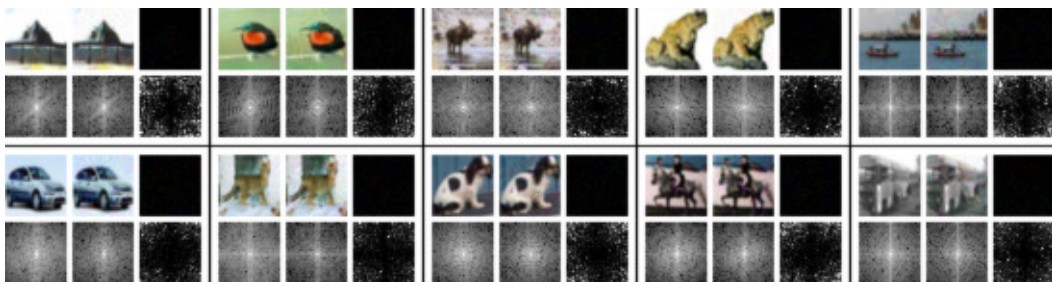

Figure 5: The log-spectrum difference of adversarial examples. In each $3 \times 2$-image part, the first line is the original example, the adversarial example attacked by PGD in a normally trained Resnet-32 and the perturbation. The second line is the log-spectrum of the original example, the log-spectrum of the adversarial example and the difference of the two log-spectrum.

## 4.3 LOW-PASS FILTER IMPROVES ROBUSTNESS

In this part, we empirically show how a low-pass filter (which filters out HFC of the inputs) can improve the robustness of the model with a cheat sheet in the sense that the attacker does not know the existence of the filter while the model is aware of the frequency distribution of the gradient-based adversarial examples.

The procedure of our **low-pass filter** basically has three steps: performing DFT on the inputs, then masking their high frequency components (set as 0) in the frequency domain and finally performing inverse DFT on them to give the desired inputs with only low frequency components preserved.

We evaluate two types of accuracy. One is filtering the clean images out of high frequencies and feed them into the model to test accuracy, named as "**no-attack accuracy**". The other is firstly PGD-attacking the input images in the normally trained model, and then check the test accuracy of filtered adversarial images, called "**PGD-attack accuracy**".

Here, we use data augmentation to train the model to the best. We show the results in Fig. 6. As the cut-off increases, the no-attack accuracy gradually increases, and the PGD-attack accuracy first increases and then decreases. The highest robustness achieves 51.7% when the cut-off $k = 12$ and its robustness accuracy is 81.7%. And the normal accuracy (no filter, no attack) is 93.03%. We also adversarially train the model with PGD-attacks. Its normal accuracy is 79% and PGD-attack

accuracy is 48% without the filtering. Our low-pass filter model not only performs better on both no-attack accuracy and PGD-attack accuracy, but also does not need computationally expensive adversarial training.

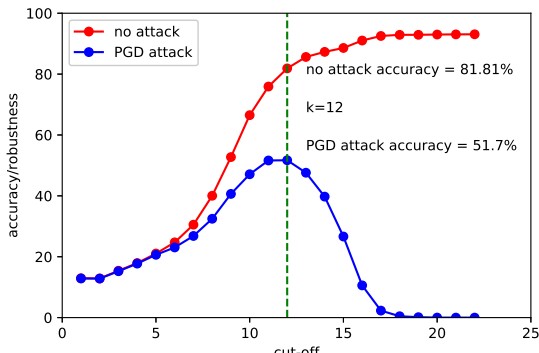

Figure 6: Normally trained model's performance and robustness after a low-pass filter with cut-off from 1 to 22.

Theorem 3 provides a reasonable explanation for the above results. When the cut-off is low, the filter masks most of the original image and the adversarial perturbation, so both of the no-attack and PGD-attack models perform not well. When the cut-off is proper, there remains enough information of original images but little of perturbations, so the PGD attacker cannot fool the model thoroughly. Therefore, with a suitable cut-off, the low-pass filter can indeed improve the model's robustness without computationally expensive adversarial training.

## 5 CONCLUSION

We investigate the adversarial examples through the lens of frequency analysis. Our work both theoretically and empirically clarifies the definition of log-spectrum difference of the adversarial examples in existing literature. Besides, we devote to understanding the spectral trajectories of adversarial examples: $l_2$ FGM perturbations gradually increase the concentration in the low-frequency part of the spectrum but their ratios of LFC will never exceed that of clean data during the training process over model parameters. Inspired by these findings, we find that a low-pass filter can improve the robustness of the model and give a reasonable explanation for this phenomenon.

Future work can focus on analysis of other perturbations in the frequency domain such as FGSM and $l_\infty$ PGD perturbations or providing theoretical understanding of the phenomenon that the ratio of the LFC of the perturbation for adversarially trained model is much higher than that for normally trained model (Appendix D).

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

## A  PROOFS

### A.1  PROOF OF THEOREM 1

According to (8), the update of $\tau(\tilde{\boldsymbol{\delta}}[k])$ for $k$-th frequency component of $l_2$ norm FGM adversarial perturbation at the $(t+1)$-th step is proportional to

$$(1 - 2\bar{\eta}_2^{(t)})|\widetilde{\nabla f}^{(t)}[k]|^2 - 2\tilde{\eta}^{(t)}|\widetilde{\nabla f}^{(t)}[k]||\tilde{\boldsymbol{x}}[k]|\cos\left(\triangle\varphi_k^{(t)}\right). \tag{13}$$

We adopt the following representations to see trends of $\cos\left(\triangle\varphi_k^{(t)}\right)$ through the training process:

$$\widetilde{\nabla f}^{(t)}[k] := \overrightarrow{c_{(t)}} = (|c_{(t)}|\cos\theta, |c_{(t)}|\sin\theta)$$
$$\tilde{\boldsymbol{x}}[k] := \overrightarrow{x_{(k)}} = (|x_{(k)}|\cos\zeta, |x_{(k)}|\sin\zeta),$$

then $|\widetilde{\nabla f}^{(t+1)}[k]||\tilde{\boldsymbol{x}}[k]|\cos\left(\triangle\varphi_k^{(t)}\right)$ will have the form of

$$\left\langle \overrightarrow{c_{(t+1)}}, \overrightarrow{x_{(k)}} \right\rangle = (1 - \bar{\eta}^{(t)})\left\langle \overrightarrow{c_{(t)}}, \overrightarrow{x_{(k)}} \right\rangle - \tilde{\eta}^{(t)}|x_{(k)}|^2 + \mathcal{O}(\eta^2)$$

at the $(t+1)$-th step. At the $(t+1)$-th step, we have

$$\triangle\tau_l^{(t+1)} - \triangle\tau_h^{(t+1)} = (1 - \bar{\eta}^{(t)})(\triangle\tau_l^{(t)} - \triangle\tau_h^{(t)}) + \tilde{\eta}^{(t)}(L_x - H_x), \tag{14}$$

which means that if $\triangle\tau_l^{(t)} > \triangle\tau_h^{(t)}$ then we must have $\triangle\tau_l^{(t')} > \triangle\tau_h^{(t')}$ for all $t' \geq t$. Besides, there must be a step which leads to this condition since $L_x > H_x$. Let $L^{(t)} = \sum_{k=0}^{k_c}|\widetilde{\nabla f}[k]|^2$ and $H^{(t)} = \sum_{k=k_c+1}^{(d-1)/2}|\widetilde{\nabla f}[k]|^2$. At the $(t+1)$-th step, the changed amount of $\tau(\tilde{\boldsymbol{\delta}} \in S_l)$ is

$$\frac{(1 - 2\bar{\eta}^{(t)})L^{(t)} + \tilde{\eta}^{(t)}\triangle\tau_l^{(t)}}{(1 - 2\bar{\eta}^{(t)})(L^{(t)} + H^{(t)}) + \tilde{\eta}^{(t)}(\triangle\tau_l^{(t)} + \triangle\tau_h^{(t)})} - \frac{L^{(t)}}{L^{(t)} + H^{(t)}}$$
$$= \tilde{\eta}^{(t)}\frac{\triangle\tau_l^{(t)}\tau\left(\tilde{\boldsymbol{\delta}}^{(t)} \in S_h\right) - \triangle\tau_h^{(t)}\tau\left(\tilde{\boldsymbol{\delta}}^{(t)} \in S_l\right)}{\sum_{k=0}^{(d-1)/2}|\widetilde{\nabla f}^{(t+1)}[k]|^2}. \tag{15}$$

If $t > t_0$ and $\tau\left(\tilde{\boldsymbol{\delta}}^{(t)} \in S_h\right) > \tau\left(\tilde{\boldsymbol{\delta}}^{(t)} \in S_l\right)$ then the above changed amount will be positive and gradient descent will increase $\tau\left(\tilde{\boldsymbol{\delta}} \in S_l\right)$ at this step. If $\triangle\tau_l^{(t)}\tau\left(\tilde{\boldsymbol{\delta}}^{(t)} \in S_h\right) < \triangle\tau_h^{(t)}\tau\left(\tilde{\boldsymbol{\delta}}^{(t)} \in S_l\right)$, then gradient descent will increase $\tau\left(\tilde{\boldsymbol{\delta}}^{(t)} \in S_h\right)$ untill the step $t_1$ at which $\triangle\tau_l^{(t_1)}\tau\left(\tilde{\boldsymbol{\delta}}^{(t_1)} \in S_h\right) \geq \triangle\tau_h^{(t_1)}\tau\left(\tilde{\boldsymbol{\delta}}^{(t_1)} \in S_l\right)$, then for any $t > t_1$, we have $\tau\left(\tilde{\boldsymbol{\delta}}^{(t+1)} \in S_l\right) > \tau\left(\tilde{\boldsymbol{\delta}}^{(t)} \in S_l\right)$.

### A.2  PROOF OF THEOREM 2

We provide the proof for ReLU activation function for positive $\tilde{eta}a$, the case for negative $\tilde{\eta}$ is similar. The update rules of $R_k$ at the $(t+1)$-th step are

$$R_k^{(t+1)2} = R_k^{(t)2} - 2\tilde{\eta}^{(t)}R_k^{(t)}\cos\left(\triangle\varphi_k^{(t)}\right)$$
$$R_k^{(t)}\cos\left(\triangle\varphi_k^{(t)}\right) = R_k^{(t-1)}\cos\left(\triangle\varphi_k^{(t-1)}\right) - \tilde{\eta}^{(t-1)}.$$

For any $k > k' > 0$ with $R_k^{(0)} > R_{k'}^{(0)}$, we have

$$R_k^{(t+1)2} - R_{k'}^{(t+1)2} = R_k^{(t)2} - R_{k'}^{(t)2} - 2\tilde{\eta}^{(t)}\left[R_k^{(t)}\cos\left(\triangle\varphi_k^{(t)}\right) - R_{k'}^{(t)}\cos\left(\triangle\varphi_{k'}^{(t)}\right)\right]$$
$$= R_k^{(0)2} - R_{k'}^{(0)2} - 2\sum_{t'=0}^{t}\tilde{\eta}^{(t')}\left[R_k^{(0)}\cos\left(\triangle\varphi_k^{(0)}\right) - R_{k'}^{(0)}\cos\left(\triangle\varphi_{k'}^{(0)}\right)\right]$$

at the $(t+1)$-th step. If $R_k^{(0)} \cos\left(\triangle\varphi_k^{(0)}\right) - R_{k'}^{(0)} \cos\left(\triangle\varphi_{k'}^{(0)}\right) > 0$, considering the case $\tilde{\eta}^{(t')} = \tilde{\eta}_{\max}$ for any $t'$ and the condition that the left L.H.S of the above equation is larger than $0$ gives us Eq.(11). On the other hand, if $R_k^{(0)} \cos\left(\triangle\varphi_k^{(0)}\right) - R_{k'}^{(0)} \cos\left(\triangle\varphi_{k'}^{(0)}\right) < 0$, then $R_k^{(t)} > R_{k'}^{(t)}$ for all $t \geq 0$.

### A.3 PROOF OF THEOREM 3

We consider the case for $l_2$ FGM perturbations with $\tilde{\eta} > 0$, the case for negative $\tilde{\eta}$ is similar. Let the initialization of the network always satisfy the condition

$$\frac{L^{(0)}}{L^{(0)} + H^{(0)}} = \tau(\tilde{\boldsymbol{\delta}}^{(0)} \in S_l) < \tau(\tilde{\boldsymbol{x}} \in S_l) = \frac{L_x}{L_x + H_x} \tag{16}$$

such that $\tau(\tilde{\boldsymbol{x}} \in S_l) - \tau(\tilde{\boldsymbol{\delta}}^{(0)} \in S_l) = \zeta > 0$. Then at the $(t)$-th step, we can express $\tau(\tilde{\boldsymbol{\delta}}^{(0)} \in S_l)$ as follows

$$\frac{L^{(t)}}{L^{(t)} + H^{(t)}} = \frac{L^{(0)} + \sum_{t'=0}^{t-1} \tilde{\eta}^{(t')} \triangle\tau_l^{(t')}}{L^{(0)} + H^{(0)} + \sum_{t'=0}^{t-1} \tilde{\eta}^{(t')}(\triangle\tau_h^{(t')} + \triangle\tau_l^{(t')})}$$

$$= \frac{L^{(0)} + \sum_{t'=0}^{t-1} \tilde{\eta}^{(t')}(\sum_{t''=0}^{t'} \tilde{\eta}^{(t'')} L_x + \triangle\tau_l^{(0)})}{L^{(0)} + H^{(0)} + \sum_{t'=0}^{t-1} \tilde{\eta}^{(t')}(\sum_{t''=0}^{t'} \tilde{\eta}^{(t'')}(L_x + H_x) + \triangle\tau_l^{(0)} + \triangle\tau_h^{(0)})}.$$

Let

$$a = \sum_{t'=0}^{t-1} \tilde{\eta}^{(t')} \sum_{t''=0}^{t'} \tilde{\eta}^{(t'')} L_x, b = \sum_{t'=0}^{t-1} \tilde{\eta}^{(t')} \sum_{t''=0}^{t'} \tilde{\eta}^{(t'')} H_x, c = \sum_{t'=0}^{t-1} \tilde{\eta}^{(t')} \triangle\tau_l^{(0)}, d = \sum_{t'=0}^{t-1} \tilde{\eta}^{(t')} \triangle\tau_h^{(0)},$$

then

$$\frac{a}{a+b} = \tau(\tilde{\boldsymbol{x}} \in S_l), \frac{c}{c+d} = \frac{\triangle\tau_l^{(0)}}{\triangle\tau_l^{(0)} + \triangle\tau_h^{(0)}}$$

and

$$\frac{L^{(t)}}{L^{(t)} + H^{(t)}} = \frac{L^{(0)} + a + c}{L^{(0)} + H^{(0)} + a + b + c + d}$$

$$= \tau(\tilde{\boldsymbol{x}} \in S_l) \frac{1 + \frac{c}{a} + \frac{L^{(0)}}{a}}{1 + \frac{L^{(0)} + H^{(0)}}{a+b} + \frac{c+d}{a+b}}.$$

If

$$\frac{c}{a} + \frac{L^{(0)}}{a} < \frac{L^{(0)} + H^{(0)}}{a+b} + \frac{c+d}{a+b}$$

$$\frac{c}{L^{(0)} + H^{(0)}} + \tau(\tilde{\boldsymbol{\delta}}^{(0)} \in S_l) < \tau(\tilde{\boldsymbol{x}} \in S_l)\left(1 + \frac{c+d}{L^{(0)} + H^{(0)}}\right)$$

$$\triangle\tau_l^{(0)} - \tau(\tilde{\boldsymbol{x}} \in S_l)(\triangle\tau_l^{(0)} + \triangle\tau_h^{(0)}) < \zeta\left(L^{(0)} + H^{(0)}\right)$$

then we must have

$$\tau(\tilde{\boldsymbol{\delta}}^{(t)} \in S_l) < \tau(\tilde{\boldsymbol{x}} \in S_l) \tag{17}$$

at the $t$-th step. A stronger condition is that

$$\triangle\tau_l^{(0)} < \tau(\tilde{\boldsymbol{x}} \in S_l)(\triangle\tau_l^{(0)} + \triangle\tau_h^{(0)})$$

then we will have $\tau(\tilde{\boldsymbol{\delta}}^{(t)} \in S_l) < \tau(\tilde{\boldsymbol{x}} \in S_l)$ for all $t$. This condition approximately states that the "ratio" of low frequency component of $\triangle\tau_l^{(0)} + \triangle\tau_h^{(0)}$ is smaller than that of clean data.

# B $l_2$ PGD PERTURBATIONS

The PGD update rule for finding perturbations of $x$ with learning rate $\xi$ at step $j + 1$ is:

$$\boldsymbol{\delta}^{(j+1)} = \mathcal{P}_{\mathcal{B}(0,\epsilon)}\left[\boldsymbol{\delta}^{(j)} + \xi\frac{\partial \ell}{\partial f}^{(j)}\nabla_x f^{(j)}\right], \tag{18}$$

where $\mathcal{B}(0, \epsilon)$ is a ball centered at 0 with radius $\epsilon$ in Euclidean space and $\mathcal{P}$ is the projection operator defined as

$$\mathcal{P}_{\mathcal{B}(0,\epsilon)}[\boldsymbol{\delta}] = \underset{\boldsymbol{\delta}' \in \mathcal{B}(0,\epsilon)}{\operatorname{argmin}} \|\boldsymbol{\delta}' - \boldsymbol{\delta}\|^2.$$

Note that in this part quantities removing the step script $j$ (e.g. $\frac{\partial \ell}{\partial f}$ and $\nabla_x f$) refers to not involving perturbations. For convenience, we use the same learning rate $\xi$ for all steps $j$ and instead explore the update of

$$\boldsymbol{\kappa}^{(j)} = \frac{\boldsymbol{\delta}^{(j)}}{\xi} \tag{19}$$

to the order of $\delta$ to see trends of PGD perturbations in frequency domain alongside PGD iterations since $\tau(\tilde{\boldsymbol{\delta}}[k]) = \tau(\tilde{\boldsymbol{\kappa}}[k])$. For ease of notation, we adopt the following representations: Let $\triangle \phi_k^{(j)}$ denote the difference of phases between $\widetilde{\nabla f}[k]$ and $\tilde{\boldsymbol{\kappa}}^{(j)}[k]$ ; let $\bar{\beta}^{(j)} = \partial \ell / \partial f + \boldsymbol{\delta}^{(j)} \cdot \nabla_x f$ where $\bar{\beta}^{(0)} > 0$ and one can drive similar results for $\bar{\beta}^{(0)} < 0$; we denote

$$\triangle \bar{\tau}_l^{(j)} \triangleq 2\sum_{k=0}^{k_c} |\tilde{\boldsymbol{\kappa}}^{(j)}[k]||\widetilde{\nabla f}[k]|\cos\left(\triangle\phi_k^{(j)}\right) \text{ and } \triangle\bar{\tau}_h^{(j)} \triangleq 2\sum_{k=k_c+1}^{(d-1)/2}|\tilde{\boldsymbol{\kappa}}^{(j)}[k]||\widetilde{\nabla f}[k]|\cos\left(\triangle\phi_k^{(j)}\right),$$

where $\bar{\beta}^{(j)}\triangle\bar{\tau}_l^{(j)}$ and $\bar{\beta}^{(j)}\triangle\bar{\tau}_h^{(j)}$ are changed amounts of LFC and HFC of $|\kappa|^2$ at the $(j + 1)$-th step of PGD iteration. We provide below our results on frequency spectrum of $l_2$-norm PGD perturbations.

**Theorem 4 (The spectral trajectory of $l_2$ PGD perturbation)** *At the $(j + 1)$-th step of PGD, iteration of PGD will change the ratio of LFC of $l_2$ norm PGD adversarial perturbation for a neural network (1) which satisfies $|\partial \ell / \partial f| = \epsilon^{1+\nu}$ with $0 < \nu < 1$ as follows,*

$$\tau(\tilde{\boldsymbol{\delta}} \in S_l) \leftarrow \tau(\tilde{\boldsymbol{\delta}} \in S_l) + \bar{\beta}^{(j)}\frac{\triangle\bar{\tau}_l^{(j)}\tau\left(\tilde{\boldsymbol{\delta}}^{(j)} \in S_h\right) - \triangle\bar{\tau}_h^{(j)}\tau\left(\tilde{\boldsymbol{\delta}}^{(j)} \in S_l\right)}{\sum_{k=0}^{(d-1)/2}|\tilde{\boldsymbol{\kappa}}^{(j+1)}[k]|^2}. \tag{20}$$

**Remark** If the two-layer neural network in (1) is trained with at least $t \geq t_1$ steps ($t_1$ determined by theorem 1) such that $\tau(\widetilde{\nabla f} \in S_l) > \tau(\widetilde{\nabla f} \in S_h)$, then, according to theorem 4, there exists

$$j_0 = \max\left\{0, \frac{\triangle\bar{\tau}_h^{(0)} - \triangle\bar{\tau}_l^{(0)}}{\tilde{\beta}\left(\sum_{k=0}^{k=k_c}|\widetilde{\nabla f}[k]|^2 - \sum_{k=k_c}^{k=(d-1)/2}|\widetilde{\nabla f}[k]|^2\right)}\right\}, \tag{21}$$

where $\tilde{\beta} = \max_{i \in [0,j_0]}\bar{\beta}^{(i)}$, such that $\tau(\tilde{\boldsymbol{\delta}}^{(j+1)} \in S_l) > \tau(\tilde{\boldsymbol{\delta}}^{(j)} \in S_l)$ for all $j > j_0$ if $\tau(\tilde{\boldsymbol{\delta}}^{(j)} \in S_l) < \tau(\tilde{\boldsymbol{\delta}}^{(j)} \in S_h)$.

## B.1 PROOF OF THEOREM 4

At the (j + 1)-th step of PGD update for $\boldsymbol{\kappa}^{(j+1)}$, if

1. $\mathcal{P}_{\mathcal{B}(0,\epsilon)}^{(j)} = I$:

$$\boldsymbol{\kappa}^{(j+1)} = \boldsymbol{\kappa}^{(j)} + \frac{\partial \ell}{\partial f}\nabla_x f + \frac{\partial \ell}{\partial f}\sum_r a_r \sigma'' \boldsymbol{\delta}^{(j)} \cdot \boldsymbol{W}_{:,r}\boldsymbol{W}_{:,r} + \boldsymbol{\delta}^{(j)} \cdot \nabla_x f\nabla_x f + \mathcal{O}(\boldsymbol{\delta}^2);$$

2. $\mathcal{P}_{\mathcal{B}(0,\epsilon)}^{(j)} \neq I$:

$$\boldsymbol{\kappa}^{(j+1)} = \frac{\epsilon}{\|\boldsymbol{\kappa}^{(j)} + \nabla_{\boldsymbol{x}}\ell^{(j)}\|} \left( \boldsymbol{\kappa}^{(j)} + \frac{\partial\ell}{\partial f}^{(j)} \nabla_{\boldsymbol{x}} f^{(j)} \right).$$

In either case, the quantity $\tau(\tilde{\boldsymbol{\kappa}}[k])$ is proportional to

$$\left| \mathcal{F} \left( \boldsymbol{\kappa}^{(j)} + \frac{\partial\ell}{\partial f}^{(j)} \nabla_{\boldsymbol{x}} f^{(j)} \right) \right|^2 = \left| \mathcal{F} \left( \boldsymbol{\kappa}^{(j)} + \frac{\partial\ell}{\partial f} \nabla_{\boldsymbol{x}} f + \boldsymbol{\delta}^{(j)} \cdot \nabla_{\boldsymbol{x}} f \nabla_{\boldsymbol{x}} f \right) \right|^2$$

since the term $\boldsymbol{\delta}\partial\ell/\partial f < \delta^2$ and can be dropped. Therefore, we now consider

$$\tilde{\boldsymbol{\kappa}}^{(j+1)}[k] = \tilde{\boldsymbol{\kappa}}^{(j)}[k] + \left( \frac{\partial\ell}{\partial f} + \boldsymbol{\delta}^{(j)} \cdot \nabla_{\boldsymbol{x}} f \right) \widetilde{\nabla f}[k] \tag{22}$$

at the $(j+1)$-th step of PGD in the frequency domain to explore ratio of frequency $k$ to the whole frequency spectrum of perturbations

$$\tau\left( \tilde{\boldsymbol{\delta}}^{(j+1)}[k] \right) \propto |\tilde{\boldsymbol{\kappa}}^{(j)}[k]|^2 + 2\left( \beta' + \boldsymbol{\delta}^{(j)} \cdot \nabla_{\boldsymbol{x}} f \right) |\tilde{\boldsymbol{\kappa}}^{(j)}[k]| |\widetilde{\nabla f}[k]| \cos(\triangle\phi_k^{(j)}). \tag{23}$$

**Lemma 1 (Dynamics of $l_2$ Norm PGD in the Frequency Domain)** *If the initialization of $l_2$ norm PGD adversarial perturbation satisfies $\frac{\partial\ell}{\partial f} + \boldsymbol{\delta}^{(0)} \cdot \nabla_{\boldsymbol{x}} f > 0$, then it will be positive at every iteration of PGD[3].*

Similar to Eq.(15), one can derive the changed amount of $\tau(\tilde{\boldsymbol{\delta}} \in S_l)$ in theorem 4 at the $(j+1)$-th step and find the necessary condition.

## C  SUPPLEMENTARY EXPERIMENT

In this part, we present experiments on models with MSE loss and two-layer neural networks with fixed outer-layer parameters in Section 4.

### C.1  MSE LOSS

We show the supplementary experiment of Section 4.1 in Fig. 7 and Fig. 8. They have similar results to show that the spectrum of adversarial perturbations are more concentrated in the low-frequency domain after a sufficient training.

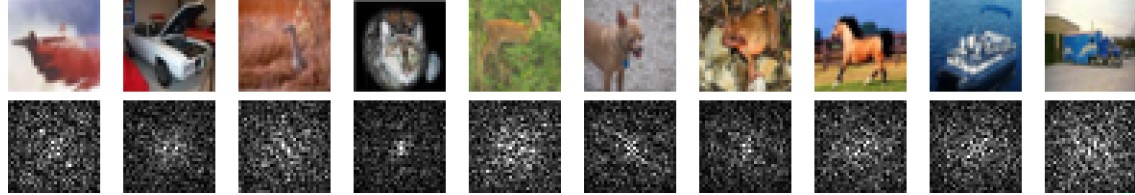

Figure 7: The difference of the spectrum between original and adversarial examples. The model are trained with MSE loss.

Then we show the supplementary experiment of Section 4.2 in Fig. 9 and Fig. 10. They have similar results to show the log-spectrum difference of adversarial examples is generally concentrated around.

In general, there is little empirical gap between MSE loss and CrossEntropy loss.

---

[3]A similar conclusion exists when $\frac{\partial\ell}{\partial f} + \boldsymbol{\delta}^{(0)} \cdot \nabla_{\boldsymbol{x}} f < 0$.

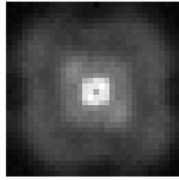

Figure 8: The expectation of the spectrum of adversarial perturbations. The model are trained with MSE loss.

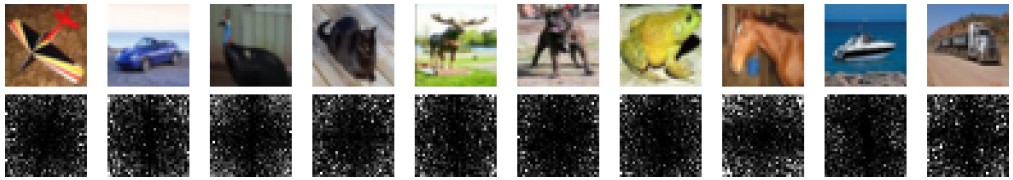

Figure 9: The difference of the log-spectrum between original and adversarial examples. The model are trained with MSE loss.



Figure 10: The expectation of the log-spectrum of adversarial perturbations. The model are trained with MSE loss.

### C.2 TWO-LAYER NEURAL NETWORK

We use two-layer neural networks with fixed outer-layer parameters. The input layer is of size $3 * 32 * 32$ followed by a ReLU activation function, and the hidden layer is of size $500$ followed by a sigmoid activation function. The dataset is still CIFAR10. We use random initialization and SGD optimizer without any training tricks. This setting can be used in practice and get higher than 40% test accuracy. The attack method is the same as the setup in Section 4.

We show the LFC ratio of the adversarial perturbations during the training process in Fig.11. After sufficient training, we show the expectation of the spectrum in Fig.12(a) and the expectation of the log-spectrum difference in Fig.12(b).

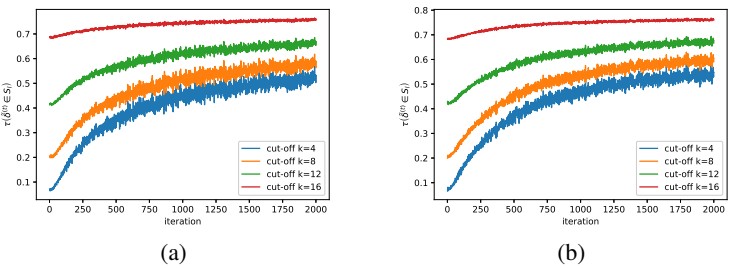

Figure 11: The LFC ratio of the adversarial perturbations for two-layer neural networks in the training process. (a) The attack method is $l_2$ FGM; (b) The attack method is $l_2$ PGD.

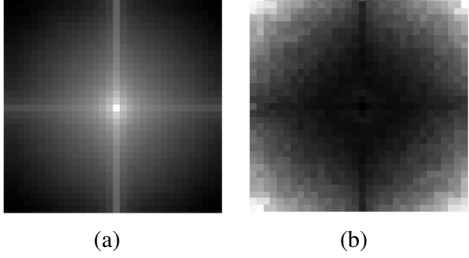

(a)                  (b)

Figure 12: For two-layer neural network, the expectation of: (a) the spectrum of adversarial perturbations; (Due to the large value gap between LFC and HFC, it is hard to see the trend in original result, so we show the logarithm of the result which will not change the size relationship.) (b) the log-spectrum difference of adversarial examples.

## D   PERTURBATIONS OF PGD-ATTACK ADVERSARIAL TRAINED MODEL ARE MORE CONCENTRATED IN LFC

We do use the same setup in Section 4, and the PGD-attack adversarial training is also PGD-attack with $\epsilon = 8/255$, 40 iterations and step size $\xi = 4/255$ in each training step. Then we test the ratio of LFC of original images, perturbations in a normally trained model and perturbations in a PGD-attack adversarially trained model. The results are shown in Fig. 13. It shows that the LFC ratio of perturbations in PGD-attack adversarially trained model is generally higher than that in normally trained model.

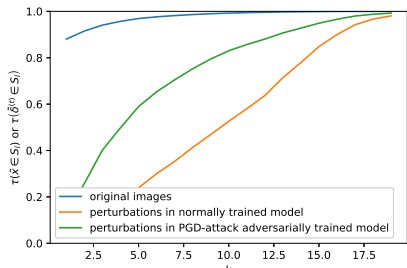

Figure 13: the ratio of LFC of original images, perturbations in a normally trained model and perturbations in a PGD-attack adversarially trained model

