# OpenReview forum: "A frequency domain analysis of gradient-based adversarial examples"
_ICLR.cc/2021/Conference — Reject_

### Official Review · AnonReviewer4 · 2020-10-20
**A frequency domain analysis of gradient-based adversarial examples**

**Rating:** 3
**Confidence:** 5

**Review:**

The paper shows formally (under somewhat restricted conditions, like 2-layer NN and for "natural images") that adversarial perturbations mainly affect high-frequency content of the images. This is also shown empirically, and it is shown that low-pass filtering improves robustness against adversarial examples.

The core idea of the paper (low-pass filtering improves robustness against AEs) is actually old. Several methods have been proposed that in one way or another perform a low-pass filtering of the image to reduce the perturbation noise and thus improve robustness to AEs. The core idea of the paper is, therefore, not novel. That perturbations must be mainly impinging on high-frequency content is an obvious conclusion. The effect of low-pass filtering on clean image classification is also, once again, neglected. What is the value of a defense method if it decreases accuracy for clean images?

Besides, the experiments are rather limited in only using CIFAR-10 (which is a subset of CIFAR) and PGD attack. The experiments are also limited insofar as a comparison is made with adversarial training, without consideration for any of the dozens other defense methods available.

Minor: On p. 1 there is a sentence "...that human can only perceive LFC." that sounds too categorical, it has to be rephrased.

---

> ### Author Response · Authors · 2020-11-14
> **Response to Reviewer #4**
>
> Thank you for your reviews. Unfortunately there are some misunderstandings on our main contributions and we hope we can clarify them here.
>
> "adversarial perturbations mainly affect high-frequency content of the images "
> -> First of all, we did not make such a claim. On the contrary, we formally showed that the gradient-based adversaries actually concentrate more in the low-frequency domain gradually during the normal training.
>
>  "The core idea of the paper is actually old."
> -> The core contribution of our paper is, as we showed, theoretical analysis of gradient-based adversarial perturbations in the frequency domain. It is actually a misunderstanding that "perturbations must be mainly impinging on high-frequency content". Therefore, our claim--low-pass filter improves robustness against adversarial examples--is not an obvious conclusion and our theories can explain it.
>
> "What is the value of a defense method if it decreases accuracy for clean images?"
> -> [1] discovered the trade-off between robustness and standard accuracy and many defense methods indeed cause clean accuracy decrease. There are also a series of works [2][3] devoting to understanding and explaining this phenonmenon. Decrease of clean accuracy did not make these defense methods valueless to the researcommunity.
>
> "the experiments are rather limited in only using CIFAR-10 (which is a subset of CIFAR) and PGD attack."
> -> CIFAR-10 meets our assumption of natural images which is enough and appropriate for our work. The reason for using PGD-attack is because our conclusion is based on gradient-based adversarial attacks.
>
> As for the minor point, the claim of the referred sentence is made by [4] which we cite in the article, and they gave sufficient experiments to support it.
>
> [1] Madry et al. "Robustness May Be at Odds with Accuracy"
> [2] Hongyang Zhang et al. "Theoretically Principled Trade-off between Robustness and Accuracy".
> [3] Aditi Raghunathan et al. "Adversarial Training Can Hurt Generalization".
> [4] Haohan Wang et al. "High-frequency Component Helps Explain the Generalization of Convolutional Neural Networks".

---

### Official Review · AnonReviewer1 · 2020-10-27
**Interesting results but the paper needs improvement**

**Rating:** 4
**Confidence:** 4

**Review:**

This paper analyzes the frequency spectrum of adversarial perturbations during normal training. The authors show that the low frequency component (LFC) of adversarial perturbation is increasing during training, but it is not increasing fast enough, so the LFC of adversarial perturbation is not as dense as the input natural images which obey the power law. Therefore, the log-spectrum difference of adversarial examples will express a HFC-concentrated phenomenon. The authors used theory and experiments to demonstrate this point.

The theoretical analyses look correct to me and they look useful since rigorous understanding of frequency spectrum is useful to the robustness research community. However, I have the following concerns:

1) This paper analyzes the frequency spectrum of normal training. Thus it aims to understand the adversarial perturbation for naturally-trained (i.e., non-robust model). The authors indeed provide an iterative equation showing how the frequency pattern changes across training, but it is only for a very specific case: the network is two-layer, trained with gradient descent algorithm, and the adversarial perturbation is FGM. As we know, the naturally trained models are very non-robust, which means that there are potentially many different ways to attack the model. If we constrain the attack direction to low/high frequency matrices, we may always be able to find a direction to attack the model. In fact, Ford et al (Adversarial Examples Are a Natural Consequence of Test Error in Noise) show that the existence of adversarial example is equivalent to test error under Gaussian noise, and therefore an adversarial perturbation that has uniform Fourier spectrum may be a valid perturbation direction (although it is not necessarily a direction computed from FGM). Therefore, I don't think the theory and experimental observation can be applied in a wide range of problems.

2) I think the paper is not written very clearly. When I read the second paragraph on page 2, I found the three main contributions kind of contradicting with each other. The first point says the adversarial perturbation concentrates on the low frequency domain and the second point says it concentrates on the high frequency domain. I was only able to roughly understand it when reading page 6, the paragraph after Theorem 2. I think the authors should spend more effort to make their main point clearly delivered.

3) I think the most misleading part in this paper is the claim that low pass filter improves robustness. This experiment was done by first attack the natural images on normal models, without knowing that there will be a low pass filter, and then pass the attacked images to the low pass filter. This is not consistent with the commonly used white box attack protocol, i.e., the adversary should know all the components of the model. The authors got the results by hiding the low pass filter from the adversary, and thus I think claiming the filtering improves robustness is very misleading to the readers.

===============

After author response:
I would like to thank the authors for providing response. For the second point, I am still a bit confused about spectrum of perturbations vs log-spectrum of adversarial examples. The authors agreed that my third point was correct, i.e., the current results do not give enough insights on how to improve robustness against adversarial examples, in the real-world white box setting. Given the above reasons, I decided to keep my score.

---

> ### Author Response · Authors · 2020-11-14
> **Response to Reviewer #1**
>
> Thank you a lot for the thoughtful reviews and valuable comments. We hope to solve your concerns here.
>
> 1. Our article foucses on the theoretical understanding of the gradient-based adversarial examples in the frequency domain which is still missing in the existing literature. The setting of two-layer neural network trained with gradient descent method is a good start for this purporse: we provide a feasible way to theoretically crack the problem for general depth neural networks. For other gradient-based adversaries, we presented the results for $l_2$ PGD attack (the information of gradient of the network which is necessary in this case has been analyzed in the FGM attack) in the Appendix.
> We now provide some values of our work to the community as follows. Future work can explore other types of gradient-based adversaries, e.g. FGSM, with similar methods not only for normally trained neural networks but also for adversarially trained ones and design new defense methods against gradient-based adversaries from the perspective of frequency which will be of great importance and novel. It is also possible to formally understand adversarial training in the frequency domain following our settings.
>
> 2. Sorry for the confusion on our contributions. Perhaps this is due to our claims that perturbations concentrate in LFC while log-spectrum difference concentrates in HFC in Sec.1 P.2. We will revise the structure of the article: clarifying these two terms, spectrum of perturbations and log-spectrum of adversarial exmaples, before demonstrating their distinct conclusions to avoid possible confusion.
>
> 3. Thanks for pointing out this point. The experiment of low-pass filter improving robustness is a verification of our main result that although gradient-based adversaries concentrate in the low-frequency domain they do not concentrate as dense as the natural images. Thus it is more appropriate to say that "the filter improves model's robustness with a cheatsheet" in the white-box setting.

---

### Official Review · AnonReviewer3 · 2020-10-28

**Rating:** 5
**Confidence:** 4

**Review:**

This submission deals with understanding the gradient based adversarial examples. For this means, it analyzes the adversarial examples in the frequency domain, where it identifies that the ratio of high-frequency and low frequency parts is quite large in adversarial examples compared with the natural ones. As a result, it proposes to apply a low-pass filtering to the data that can significantly improve the model robustness. Some experiments for CIFAR-10 classification are performed to support the main conclusions.


Strong points
The frequency domain perspective to analyzing adversaries is interesting and useful.


Weak points
Several assumptions are made. In particular, the assumption about the weights {a_r} being random is unrealistic.
It is not clear how much the accuracy can deteriorate because of low pass filtering. It would be important to show that reducing the cut-off frequency to the desired value that rejects the adversaries is not going to significantly hurt the accuracy.
The theoretical results and analysis relies on several assumptions. It particularly considers a very simple single-layer neural network, where the weights for the classification weights are assumed random and not learnable. The theoretical perspective is of high importance. However, the tightness of the final thresholds in Theorems 1 and 2 are not clear. There is no empirical evidence designed specifically to support the Theorems.


Suggestions
Specific experiments to support the models and assumption and results in Theorem 1 and 2 would increase the value of this work.


Reproducibility
The code is not provided, but the details of the experiments are listed.

---

> ### Author Response · Authors · 2020-11-14
> **Response to Reviewer #3**
>
> Thanks a lot for your thoughtful reviews and useful suggestions.
>
> The two-layer setting with fixing out-layer parameters is unrealistic: Theoretical analysis on deep neural network is definitely important but extremely challenging. In fact, our work on two-layer nerual network is the first step towards this direction--theoretically understanding the gradient-based adversaries themselves in the frequency domain. And our theoretical findingings are supported by experiments on neural networks with more complicated architectures (ResNet-32): in Fig 2(a), the ratios of LFC of $l_2$ FGM perturbations gradually increase during the normal training as claimed by Theorem 1; Fig 6(b) indicates that the expectation of log-spectrum difference of adversdarial examples indeed expresses a high-frequency concentration which is demonstrated in Theorem 2. The assumption of weights $a_r$ being not learnable has been widely used in earlier works of theoretical analysis for neural networks e.g. [1]. To better support these assumptions and results in Theorem 1 and Theorem 2, we will supplement experiments on a two-layer neural network model with fixed weights $a_r$.
>
> It is important to show that low-pass filter will not significantly hurt the accuracy: Considering the dense-concentration in low frequency of natural images, the model would have surprisingly high accuracy with significantly low cuf-off frequency as showed by [2] and our experiment where the model achieves 82% clean accuracy with the cut-off chosen as approximately half of the maximum frequency (Fig. 3). Therefore, setting the cut-off frequency to the desired value will not hurt the accuracy much while rejecting the adversarial perturbations a lot. However, we can not provide the theoretical guarantees on the exact clean accuracy of the model preprocessed by low-pass filtering with a particular cut-off frequency.
>
> [1] Sanjeev Arora et al. "Fine-Grained Analysis of Optimization and Generalization for Overparameterized Two-Layer Neural Networks".
> [2] Haohan Wang et al. "High-frequency Component Helps Explain the Generalization of Convolutional Neural Networks".

---

### Official Review · AnonReviewer2 · 2020-11-01
**The paper proposes that applying a low pass filter on the input data improves the robustness of the model.**

**Rating:** 7
**Confidence:** 4

**Review:**

The paper proposes that applying a low pass filter on the input data improves the robustness of the model.
•	Filtering out the high frequency component of the signal can be very application dependent. In many applications, e.g. Time-frequency image of the signals such as radar signals or vibration signals, the high frequency component contains the information that the modeler of the DNN is seeking to model. Therefore, the authors should specify the applications in which the filtering is actually useful. Please elaborate more in what application this technique can be useful. Image processing is a very general term that may not be necessarily descriptive about the domain where the proposed technique is applicable.
•	Can the authors specify in the paper if the frequency dependence of the DNN depends on the structure (length and width) of the designed DNN? Is there recommendations for modelers?
•	Can the authors provide more details on the design of the low pass filter? Is it FIR or IIR? How do they come up with the design of the filter?

---

> ### Author Response · Authors · 2020-11-15
> **Response to Reviewer #2**
>
> Thank you a lot for your thoughtful reviews and constructive comments.
>
> Specify the applications in which the filtering is actually useful: Thanks for pointing out that. Our theorems are for natural images which obey the power-law. Therefore, the application is also for natural images, e.g classification. We will make it more clear in the revision.
>
> Structure of DNN: It is indeed an insteresting problem. Two-layer neural network and Resnet are two benchmarks of deep learning models in theoretical and empirical anylysis. We developed theory on two-layer networks with non-linear activation function, and the experimental results on both two-layer networks and Resnet are coincident. We empirically found that different structure of DNNs does effect the model performance in frequency domain, so as to many additional factors like dataset and training strategy. However, we can't draw a regular conclusion. In the future, we will further study it theoretically and empirically.
>
> Details on the design of the low pass filter: The procedure of our low-pass filter basically has three steps: performing DFT on the inputs, then masking their high frequency components (set as 0) in the frequency domain and finally performing inverse DFT on them to give the desired inputs with only low frequency components preserved. We will show such process in the revision.

---

### Public Comment · ~Guillermo_Ortiz-Jimenez1 · 2020-11-11
**Claims in submission are not applicable to other datasets**

Some of the claims of the paper might only apply to CIFAR-10. In particular, the fact that adversarial perturbations are mostly a high-frequency phenomenon is not true for ImageNet or MNIST trained CNNs.

In our recent NeurIPS paper [1] we show that the distance to the boundary (margin) in different frequency bands is heavily dependent of the distribution used for training, and that in the case of ImageNet and MNIST classifiers it is mostly low in the low frequency subspaces. Meanwhile, we also show that during adversarial training on CIFAR-10 the l2-PGD adversarial perturbations have always higher energy in the low-frequency subspaces.

We also show that training on a low-pass version of CIFAR-10 mostly affects the margin in the high frequency spectrum. Thus, the adversarial examples of a network trained on low-pass CIFAR-10 are not high-frequency. This is due to the tendency of neural networks to create invariances along subspaces where the input data has no energy. This again shows that the claim in this submission (adversarial perturbations are high frequency) is not true in general.

In another workshop paper at ICML [2], we show in fact that inducing invariance along the high frequencies using a low-pass transformation can induce robustness to certain types of distribution shifts. However, we never observed any improvement in l2-PGD adversarial robustness.

We believe that these two papers are very relevant for the study presented in this submission. In fact, we would appreciate if the authors could clarify their claims with respect to our previous analysis. Anyway, we would kindly ask the authors to reference them accordingly in their submission.

[1] G. Ortiz-Jimenez, A. Modas, SM. Moosavi-Dezfooli, P. Frossard. "Hold me tight! Influence of discriminative features on deep network boundaries".  NeurIPS 2020

[2] G. Ortiz-Jimenez, A. Modas, SM. Moosavi-Dezfooli, P. Frossard. "Redundant features can hurt robustness to distribution shifts". Uncertainty & Robustness in Deep Learning Workshop @ ICML2020.

----

P.S.: Other works had also shown that adversarial perturbations are low-frequency in MNIST and ImageNet networks.

[3] Y. Sharma, G. W. Ding, and M. A. Brubaker, “On the Effectiveness of Low Frequency Perturbations”. IJCAI 2019.

[4] Y. Tsuzuku and I. Sato, “On the Structural Sensitivity of Deep Convolutional Networks to the Directions of Fourier Basis Functions,” CVPR 2019. (plots in this paper are not fftshifted)

---

> ### Author Response · Authors · 2020-11-14
> **Response to Guillermo Ortiz-Jimenez**
>
> Thank you for your interests.
>
> Unfortunately, you might misread our conclusion. In fact, our theorems and experiments show that the adversarial perturbations are a low-frequency phenomenon on CIFAR10 in Theorem 1 and Sec.4.1 (Fig.2, Fig.4). We also conducted our experiments on MNIST and ImageNet which led to the same conclusion. Perhaps your misunderstanding comes from our experiment that low-pass filter improves robustness. We claim that the ratio of LFC of perturbations is smaller than that of original clean images (Theorem 1 and Sec.4.3), which does not contradict with the fact that adversarial perturbations are a low-frequency phenomenon.
>
> Finally, thank you for offering these relevant works which in fact support our conclusions. We will cite them in the final version of our article.

---

> > ### Public Comment · ~Guillermo_Ortiz-Jimenez1 · 2020-11-17
> > **Thanks for the response**
> >
> > Thanks for clarifying the point that you also claim that adversarial perturbations are mostly concentrated in the low frequencies. As some reviewers pointed out, this was not clear in the first version of the paper. We have read your revised manuscript and now this point is much more clear.
> >
> > We would also like to thank you for agreeing to include the relevant works we provided. We also believe they are important prior contributions that support some claims in your work, and are therefore worth discussing.
> >
> > Regarding your claim that low-pass filtering improves robustness, we would like to point out that in our experiments we never appreciated any improvement in overall robustness. That is assuming,  as Reviewer #1 pointed out, the common white-box setting in which the attacker knows the specific defense mechanism, and hence can specifically craft adversarial perturbations in the low-frequencies. However, as we discussed in [2], testing CIFAR-10 trained networks on low-pass images does decrease accuracy significantly.
> >
> > In any case, thank you very much for the conversation.

---

### Decision · Program_Chairs · 2021-01-07
**Final Decision**

**Decision:**

Reject

**Comment:**

This work performs a frequency domain analysis on gradient-based adversarial perturbations. The authors argue that the perturbation deltas are largely concentrated in the high frequency domain and suggest a low pass filtering technique to improve the robustness of image classifiers. Reviewers raised several concerns as to how broadly applicable the given claims will hold, noting that these findings will be largely dependent on how the model was trained, what data the model is trained on, and how the adversarial perturbation will be constructed. Additionally, the proposal to apply a low pass filter to improve robustness is not new---it has been attempted in several works in the past and current consensus is this only provides improved robustness to high frequency perturbations/corruptions. As a recent example, Yin et. al. studied the robustness properties of models trained with a low pass filter and found that while robustness to high frequency noise and perturbations is improved, this procedure also degrades robustness to low frequency corruptions and perturbations. By focusing the analysis quite narrowly to restricted l2-perturbations the work is missing critical nuances that are well known in the literature studying distribution shift. This AC recommends the authors connect their analysis to the broader issue of distribution shift, for example can the theory provide understanding into how to improve robustness to both high and low frequency corruptions?